# Effect of Pressure Difference between Inner and Outer Gas Layer on Micro-Tube Deformation during Gas-Assisted Extrusion

**DOI:** 10.3390/polym14173559

**Published:** 2022-08-29

**Authors:** Bin Liu, Xingyuan Huang, Shaoyi Ren, Cheng Luo

**Affiliations:** College of Advanced Manufacturing, Nanchang University, Nanchang 330031, China

**Keywords:** micro-tube deformation, double-layer gas-assisted extrusion, pressure difference, numerical simulation

## Abstract

In the process of double-layer gas-assiste extrusion of a plastic micro-tube, the tugging effect caused by the pressure difference of the gas cushion layer inside the die has a great influence on the external dimensions of the micro-tube. Therefore, this study establishes a two-phase extrusion model based on compressible gas and incompressible melt. Ansys Polyflow finite element software was used to numerically simulate the extrusion process of the melt to analyze the effect of the gas cushion layer pressure difference on the micro-tube deformation. The research shows that the shrinkage rate of the micro-tube increases with increasing pressure of the outer cushion layer, and the degree of tube wall migration increases, too. In the process of extrusion, the first normal stress difference at the entrance of the gas cushion layer shows a significant effect on the melt velocity field distribution and the extruded micro-tube cross-sectional deformation.

## 1. Introduction

Plastic micro-tubes have high added value [1] and a wide range of applications in industries, such as healthcare, communications, oil, and automotive electronics [2]. However, micro-tubes are prone to dimensional instability during the production process. In order to improve product quality and reduce waste rate, plastic micro-tube processing technology must be solved. Die design is the key to ensure the quality of plastic tubes. In the past, many scholars have conducted a lot of research on the optimal design of traditional extrusion plastic micro-tube molds [3,4,5,6,7]. Jin [3] analyzed the effect of single-cavity micro-tube extrusion die parameters on the size of polypropylene (PP) micro-tubes by an experimental method, showing that the diameter and wall thickness of micro-tubes are significantly influenced by the size of the sizing section of the die. Many researchers have studied the effects of micro-tube extrusion process parameters [8,9,10,11,12,13] on polymer rheological properties [14,15,16,17,18,19,20] and extrusion swelling by numerical analysis [21,22,23,24,25,26,27]. However, during the traditional extrusion molding process, the melt inside the die recovers elastically via wall shearing, which can easily cause defects such as extrusion swelling, distortion, and melt fracture [28,29,30,31].

The application of gas-assisted technology in plastic micro-tube extrusion can make the melt in the die change from sticky flow to slip flow, which can effectively eliminate these problems [32,33,34,35,36]. Gas-assisted extrusion technology can significantly improve the performance of the die and the quality of the product. However, the mechanism of gas-assisted extrusion is currently unclear, which leads to some limitations in its practical application. Liang [37] first proposed a gas-assisted technique based on the slit-inlet method, which formed a stable gas cushion between the polymer melt and the die, thus achieving a significant reduction in die resistance. However, this theory failed to consider the variation of flow field and rheological behavior of the melt under gas. Ren [38,39] presented a numerical model for gas-assisted extrusion of polymers with two-phase flow of gas and melt. The numerical analysis and experimental results demonstrated a significant effect of gas pressure on the shrinkage of the melt morphology. Then, the effect of gas pressure on the rheological properties of the melt was analyzed and a method for establishing a stable gas cushion layer during the double-layer gas-assisted extrusion of plastic micro-tubes was obtained by Liu [40,41].

Despite the small diameter and wall thickness of the micro-tube and the high precision required [42], few studies have been conducted on the variation of pressure parameters in the inner and outer gas cushion layers of the extruded micro-tube during previous numerical simulations of double gas-assisted extrusion. In addition, there is a lack of theoretical models that can be used to analyze the interaction between the inner and outer gas cushion layers and the internal stress distribution. These problems have restricted the development of micro-tube extrusion processing technology. In this paper, a geometric model of an extruded plastic micro-tube assisted by inner and outer double cushion layers is proposed based on compressible gas and incompressible melt. Numerical simulation and experimental verification of the melt flow viscoelasticity inside the extrusion die are carried out to obtain the variation of the pressure difference between the inner and outer gas cushion layers of the extruded micro-tube.

## 2. Modelling and Numerical Methods

### 2.1. Geometric and Finite Element Models

The flow process of melt in the die is shown in Figure 1a. In the process of double-layer gas-assisted extrusion of plastic micro-tubes, a stable inner cushion layer was established between the annular melt and the mandrel, and a stable outer cushion layer was established between the annular melt and the extrusion die. Due to the axisymmetric structure of the plastic micro-tubes, a simplified two-dimensional axisymmetric model was used to simulate and analyze them. In Figure 1b, ABEF represents the outer gas cushion layer, BCFG represents the melt area inside the die, FGIJ represents the melt area outside the die, CDGH represents the inner gas cushion layer, KJ represents the centerline of the micro-tube, AE represents the inner wall surface of the die, DH represents the outer wall surface of the mandrel, BF represents the interface between the outer gas cushion layer and the outer wall of the plastic micro-tube, and CG represents the interface between the inner gas cushion layer and the inner wall of the plastic micro-tube. The thickness of gas cushion layer was set as 0.2 mm, the length of gas cushion layer was set as 10 mm, the outer radius of the melt was set as 1.5 mm, the wall thickness of the melt was set as 0.5 mm, the inner radius of the melt was set as 1 mm, and the length of the melt outside the die was set as 10 mm. To improve the calculation accuracy, grid encryption was performed on the inlet and outlet of the melt, the wall of the die, and the interface between the melt and the gas cushion layer, as shown in Figure 1c. As shown in Figure 2, when the number of grids is less than 900, the calculation process is difficult to converge, and when the number of grids is greater than 3600, the micro-tube external size has little effect, but the calculation time is laborious, so 3600 is taken in the finite element model to improve the numerical accuracy and convergence speed.

### 2.2. Control and Constitutive Equations

The following assumptions were made for the finite element simulation process: due to the high viscosity and low velocity of the polymer melt and the very small viscosity and mass of the gas, the inertia forces and gravity effects on these two fluids were ignored. The gas was considered as a compressible Newtonian fluid, while the melt was considered as an incompressible non-Newtonian fluid [39]. The effects of relative slip, interpenetration, and surface tension between the gas and the melt were ignored. The simplified melt and gas control equations are as follows:

Continuity Equation:(1)$$\nabla \cdot \left(\rho_{k}\nu_{k}\right)=0$$

Momentum conservation equation:(2)$$\rho_{k}\nu_{k}\cdot \nabla \nu_{k}+\nabla p_{k}-\nabla \tau_{k}=0$$

Energy conservation equation:(3)$$\rho_{k}C_{V}\left(\frac{\partial T_{k}}{\partial t}+\nu_{k}\cdot \nabla T_{k}\right)=-\nabla \cdot q_{k}-T_{k}\left(\frac{\partial p_{k}}{\partial T}\right)\left(\nabla \cdot \nu_{k}\right)+\tau_{k}:\nabla \nu_{k}$$
where ∇ is a Hamiltonian, *ρ_k_* is the density, *v_k_* is the velocity vector, *p_k_* is the pressure, *τ_k_* is the bias stress tensor, *C_V_* is the specific heat capacity, *T_k_* is the temperature, *q_k_* is the thermal conductivity, and $\tau_{k}:\nabla \nu_{k}$ is the viscous dissipation term.

The PTT differential viscoelastic constitutive equation [43] was used to describe the flow of polymer melt as follows:(4)$$\tau_{r}=\tau_{1}+\tau_{2}$$
(5)$$\tau_{2}=2\eta_{12}D$$
(6)$$\eta_{1r}=\frac{\eta_{12}}{\eta_{1}}$$
(7)$$\exp\left[\frac{\varepsilon \lambda }{\left(1-\eta_{1r}\right)\eta_{1}}tr\left(\tau_{1}\right)\right]+\lambda \left[\left(1-\frac{\xi }{2}\right)\overset{\nabla }{\tau_{1}}+\frac{\xi }{2}\overset{\Delta }{\tau_{1}}\right]=2\left(1-\eta_{1r}\right)\eta_{1}D$$
where $\tau_{r}$ is the melt stress tensor, $\tau_{1}$ is the elastic component of the melt bias stress tensor, $\tau_{2}$ is the viscous component of the melt bias stress tensor, $\eta_{12}$ is the Newtonian viscosity component of the melt, $\eta_{1}$ is the total viscosity of the melt, $\eta_{1r}$ is the viscosity ratio of the melt, λ is the relaxation time, ε is the material intrinsic parameter for the tensile properties of the melt, ξ is the material intrinsic parameter for the shear properties of the melt, and D is deformation rate tensor.

The gas state equation was used to describe the gas flow process as follows:(8)$$P_{1}=\rho_{1}\cdot R\cdot T_{1}$$
where $P_{1}$ is the gas pressure, $\rho_{1}$ is the gas density, $T_{1}$ is the gas temperature, R is the gas constant, and R = 287 ($J\cdot kg^{-1}\cdot K^{-1}$).

### 2.3. Setting of Numerical Simulation Parameters

The equation parameters of polypropylene and auxiliary gases are shown in Table 1 [44]:

### 2.4. Boundary Conditions

The direction of melt flow is along the *z*-axis and the radial direction of the melt is along the R-axis. *f_R_* and *f_Z_* are used to represent the normal and tangential stresses at the boundary surface, respectively. The *v_R_* and *v_Z_* are used to represent the normal and tangential velocities at the boundary surface, respectively. The boundary conditions are as follows:

(1) Melt inlet boundary: BC is the melt inlet boundary. After the melt flows through the compression section and the setting section of the die, it is assumed that the melt enters the gas-assisted section as a fully developed non-Newtonian viscoelastic laminar flow. The following relations are satisfied: the melt inlet boundary is set as the flow inlet boundary, and the melt inlet flow rate is set to 20 mm^3^/s. The temperature at the melt inlet boundary is set to 200 °C. 

(2) Gas inlet boundary: AB and CD are gas inlet boundaries, which are set as pressure inlet boundaries. The inlet pressure of inner gas cushion layer is set to 7500 Pa. The inlet pressure of outer gas cushion layer is set to 5000 Pa, 6500 Pa, 7500 Pa, 8500 Pa and 10,000 Pa, respectively. The temperature of the gas inlet boundary is set to 200 °C.

(3) Wall boundary: AE and DH are wall boundaries, which are regarded as no-slip interfaces. The temperature at the wall boundary is set to 200 °C.

(4) Gas-melt interface: BF and CG are the interfaces between the gas and the melt. It is assumed that the melt and the gas do not penetrate the interface, there is no relative slip, and the forces on both sides of the interface are balanced. 

(5) Melt free surface boundary: FI and GH are the free surface boundaries of the melt. 

(6) Melt outlet boundary: IJ is the melt outlet boundary.

(7) Gas outlet boundary: EF and GH are the gas outlet boundaries.

### 2.5. Evaluation Indicators

The shrinkage rates of micro-tube dimensions were used as evaluation indicators to compare the effects of double-layer gas-assisted extrusion. The evaluation indicator formula is as follows:

*B*_0_ is the shrinkage rate of outer radius:(9)$$B_{0}=\frac{\left(1.5-R_{1}\right)}{1.5}\times 100\%$$

*B*_1_ is the shrinkage rate of inner radius:(10)$$B_{1}=\frac{\left(1-r_{1}\right)}{1}\times 100\%$$

*B*_2_ is the shrinkage rate of wall thickness:(11)$$B_{2}=\frac{\left(0.5-d_{1}\right)}{0.5}\times 100\%$$

*B*_3_ is the offset rate of micro-tube wall center:(12)$$B_{3}=\frac{\left(1.25-C_{1}\right)}{1.25}\times 100\%$$
where *R*_1_ is the outer radius of the micro-tube at the exit of the die, *r*_1_ is the outer radius of the micro-tube at the exit of the die, *d*_1_ is the wall thickness of the micro-tube at the exit of the die, and *C_1_* is the center dimension of the micro-tube wall at the exit of the die.

### 2.6. Numerical Methods

In the numerical simulation, ANSYS Workbench was used for finite element analysis, the Geometry module was used to establish the geometric models, the Mesh module was used for meshing, the POLYFLOW module was used to establish 2D axisymmetric calculation tasks and perform finite element solutions, and the CFD-Post module was used for post-processing and result analysis. Due to the rheological properties of polymer melt with high viscosity and high elasticity, the melt constitutive equation is nonlinear and the computational process is difficult to converge. To facilitate the convergence of the equation solution, the Evolution method was used to set the initial conditions of the melt inlet flow rate and relaxation time, and the Spines kinematic equation mesh optimization method was used to reset the mesh of the boundaries of the free surface.

## 3. Simulation Results and Analysis

The cross-sectional size of the micro-tube was closely related to the flow state of the melt. In the simulation described in this paper, the axial velocity field, radial velocity field, pressure field, and first normal stress difference distributions of the melt and shear rate fields under different pressure differences between the inner and outer gas cushion layers were analyzed.

### 3.1. Results and Analysis of Micro-Tube Section Size

The dimensions and shrinkage rates of the micro-tube at the exit of the die under different outer gas cushion pressures are shown in Figure 3.

As can be seen from Figure 3, when the pressure of the outer gas cushion layer increases, the shrinkage rate of the inner radius, the shrinkage rate of the outer radius and the center offset rate of the micro-tube wall all increase, but the thickness of the wall changes very little. Figure 3b shows that when the pressure of the outer gas cushion layer is 6500 Pa, the center offset rate of the micro-tube wall is close to 0, which indicates good cylindricity of the micro-tube. When the pressure increases from 5000 Pa to 1000 Pa, the slope of the wall thickness shrinkage rate is smaller, while the slope of the outer radius shrinkage rate, inner radius shrinkage rate and the center offset rate of the micro-tube wall are larger. The pressure change in the outer layer has little effect on the thickness of the wall. Increasing the pressure of the outer layer causes the melt tube wall to shrink in the direction of the inner diameter, causing the cylindricity of the micro-tube to deviate greatly and the extruded micro-tube section to appear in an irregular pattern.

### 3.2. Results and Analysis of Velocity Fields

As shown in Figure 4 and Figure 5, the melt velocity field distribution under different outer gas layers was obtained by simulation to explain the deformation caused by the flow characteristics of micro-tubes during the double-layer gas-assisted extrusion process.

From Figure 4, it can be seen that both the inner and outer surfaces of the melt at the inlet of the gas layer have a radial velocity pointing to the inside of the melt, which results in a tendency for the melt to flow to the center of the tube wall and explains the contraction of the micro-tubular wall thickness in Figure 3a. As the pressure of the outer gas cushion layer increases, the radial velocity increases on the outer surface of the melt and decreases on the inner surface, but there is always a certain radial velocity difference between the inner and outer surfaces of the melt, which leads to the migration of the center of the melt tube. Prior to the melt leaving the die, the radial velocity of the melt has been reduced to 0, which means that the velocity rearrangement of the melt in the die has been completed during the double-layer gas-assisted extrusion process. Moreover, the micro-tube section size does not change after leaving the die, which basically eliminates the problem of extrusion swelling.

From Figure 5, it can be seen that the axial velocity of the melt at the die outlet increases with the pressure of the outer gas cushion layer. The axial velocity of the melt increases gradually along the axial direction, and the axial velocity of the melt remains the same after leaving the die. This is because the gas cushion layer has a certain dragging effect on the melt, and the pressure of the gas cushion layer increases the velocity of the gas outlet, so the dragging effect also increases. As the melt inlet flow rate remains unchanged and the outlet velocity gradually increases, the melt becomes axially stretched and thinner, the melt radial cross-section shape is reconstructed, and the inner and outer diameter and wall thickness are contracted.

### 3.3. Results and Analysis of Pressure

The melt pressure field distribution at different pressures in the outer gas cushion layer during the double-layer gas-assisted extrusion process is shown in Figure 6.

It can be seen from Figure 6 that the melt produces a large pressure drop at the inlet of the gas layer and then decreases to zero in the axial direction just before the melt leaves the die front. This shows that the effect of the inner and outer gas cushion layers on melt extrusion is mainly concentrated in the front section of the die.

As can be seen from Figure 7, a large first normal stress difference occurs between the inner and outer surfaces of the melt at the inlet position of the gas layer, which is an important factor for the rearrangement of the melt velocity and pressure distribution in the die. When the pressure of the inner and outer gas layer is 7500 Pa, the first normal stress difference between the inner and outer surfaces of the melt is unevenly distributed along the radial direction. As the pressure of the outer gas layer increases, the first normal stress difference on the outer surface of the melt increases and the first normal stress difference on the inner surface decreases, which leads to a larger radial velocity on the outer surface of the melt than on the inner surface. When the pressure of the outer gas layer is 6500 Pa, the first normal stress difference between the inner and outer surfaces of the melt is similar, and the radial velocity difference between the inner and outer surfaces is smaller, as shown in Figure 4. In summary, the distribution of the first normal stress difference reflects the shrinkage rate, velocity field and pressure field of the micro-tube.

### 3.4. Results and Analysis of Shear Rate

The distribution of melt shear rate at different pressures in the outer gas cushion layer of the double-layer gas-assisted extrusion process is shown in Figure 8.

From Figure 8, it can be seen that as the pressure of the outer gas cushion layer increases, the shear rate of the inner surface of the melt decreases and the shear rate of the outer surface of the melt increases. When the pressure of the inner and outer gas cushion layers is the same, a larger shear rate is generated on the inner and outer gas cushion layer surfaces, while the shear rate at the center of the melt is very small. This indicates that the melt surface is significantly dragged by the gas cushion layer during the double-layer gas-assisted extrusion process.

## 4. Experimental Results and Analysis

As shown in Figure 9, a double-layer gas-assisted extrusion experimental platform was built to verify the effect of the gas cushion layers on the extrusion of a plastic micro-tube. The double-layer gas-assisted extrusion experimental platform includes: extruder, cooling device, traction device (GRQ-25 PVC, supplied by HUAXI Plastic Machinery Co., Ltd., Dongguan, China), gas generation system (supplied by Guangdong Air TAC Intelligent Equipment Co., Ltd., Shantou, China), gas heating control system (supplied by Shantou Shanghai Electric Machine Co., Ltd., Shanghai Extrusion), and extrusion die. The structure of the extrusion die is shown in Figure 10, and the experimental results are shown in Figure 11. The melt inlet flow rate, traction speed, and melt temperature of the double-layer gas-assisted extrusion experiment are similar to the traditional extrusion experiment, as shown in Table 2.

It can be seen from Figure 11a that during the non-gas-assisted extrusion process, there was an obvious extrusion swelling at the exit of the die and the diameter of the micro-tube gradually became smaller along the axial direction. This is due to the fact that the axial speed of the micro-tube at the outlet of the die is smaller than the traction velocity during the non-gas-assisted extrusion process, so the traction device has a dragging and pulling effect on the micro-tube. In addition, due to the tensile and shear deformation of the melt in the extrusion section during the non-gas-assisted extrusion, the cavity of the die had greater internal stress, resulting in extrusion swelling. From Figure 11b, it can be seen that when the pressure of the outer gas layer is 6500 Pa, the shape of the micro-tube at the exit position of the die is similar to that of the die cross-section, and the diameter of the micro-tube varies little along the axial direction. During the gas-assisted extrusion process, the gas cushion layer had a certain axial tug on the melt inside the die head, making the axial velocity of the micro-tube at the exit position of the die head larger than that of the non-gas-assisted extrusion, which is close to the traction velocity, thus the tugging device does not exert tugs on the micro-tubes. From Figure 11c,d, it can be seen that the outer diameter of the extruded micro-tube decreases with increasing pressure of the outer gas layer, which is consistent with the numerical simulation of the effect of pressure difference on the micro-tube section size.

## 5. Conclusions

1. The numerical simulation of two double-layer gas-assisted extrusion shows that the pressure of the gas cushion layer has a great influence on the cross-sectional size and deformation of the plastic micro-tube. Besides, with the increase of pressure difference between inner and outer gas cushion layers, the migration of micro-tube wall center changes from 1.86% to 8.46%.

2. At the entrance of the gas layer, the first normal stress difference exists inside and outside the melt and is greater than 500 pa, which causes the velocity field and stress field of the melt to rearrange. It is the fundamental reason for the shrinkage and deformation of the micro-tube at the exit position of the die.

3. In the double-layer gas-assisted extrusion process, the first normal stress difference between the inner and outer surfaces of the melt can be evenly distributed by keeping the pressure of the outer gas layer at 1000 pa lower than that of the inner gas layer, and the diameter difference between the inner and outer surfaces of the melt can be reduced, thereby reducing the size deformation of the extruded micro-tube.

## Figures and Tables

**Figure 1 polymers-14-03559-f001:**
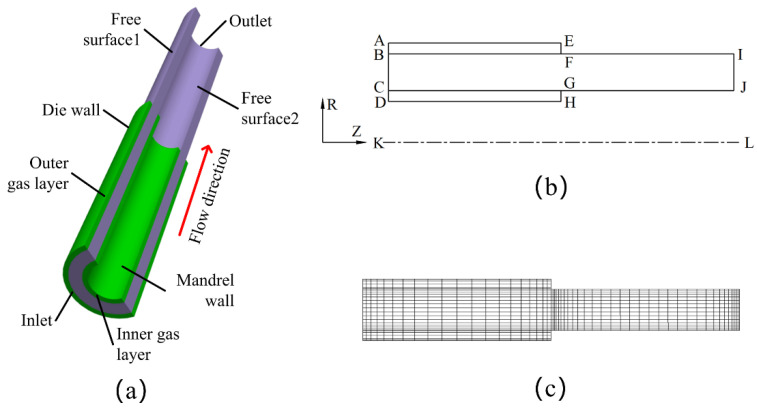
Geometric model and finite element model of the process of double-layer gas-assisted extrusion of plastic micro-tubes. (**a**) Geometric model of the flow process of melt in the die; (**b**) Simplified two-dimensional axisymmetric geometric model; (**c**) Finite element mesh model.

**Figure 2 polymers-14-03559-f002:**
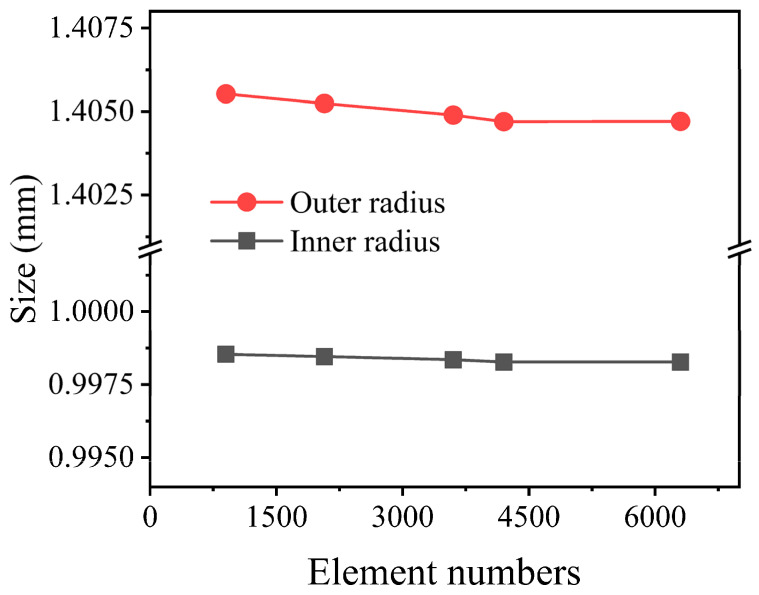
The effect of the grids number on the simulation results.

**Figure 3 polymers-14-03559-f003:**
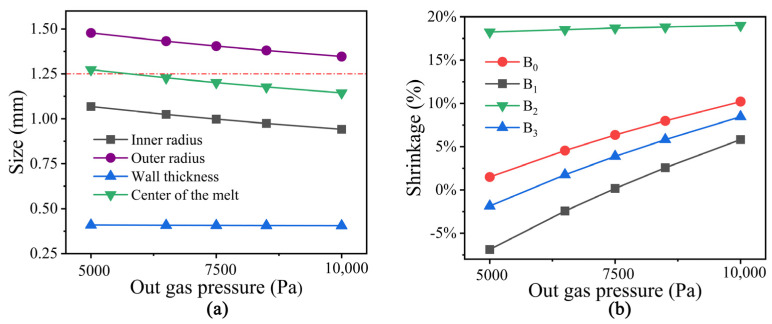
Dimensions and shrinkage rates of the micro-tube under different pressure of the outer gas cushion layer. (**a**) Dimensions of the micro-tube; (**b**) Shrinkage rate of the micro-tube.

**Figure 4 polymers-14-03559-f004:**
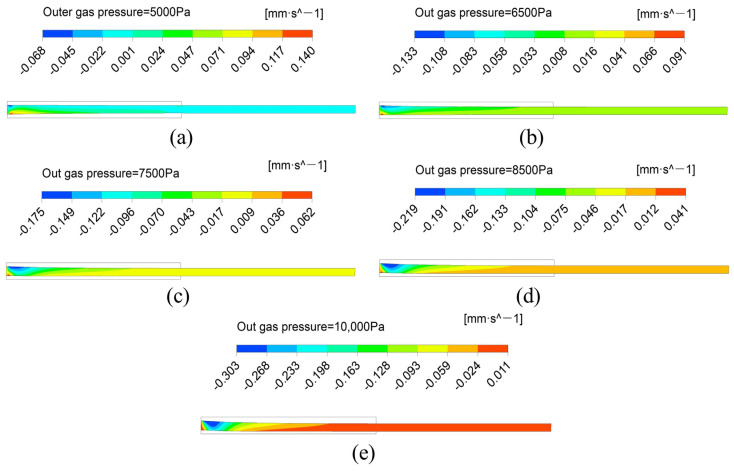
The distribution of melt radial velocity under different pressure of the outer gas cushion layer.(**a**) The pressure of the outer gas layer is 5000 Pa; (**b**) The pressure of the outer gas layer is 6500 Pa; (**c**) The pressure of the outer gas layer is 7500 Pa; (**d**) The pressure of the outer gas layer is 8500 Pa; (**e**) The pressure of the outer gas layer is 10,000 Pa.

**Figure 5 polymers-14-03559-f005:**
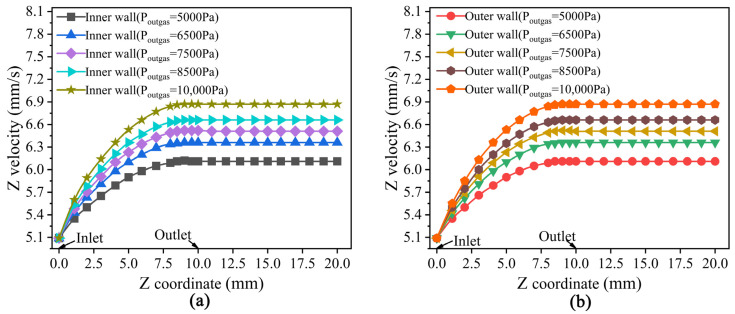
The distribution of melt Z velocity under different pressure of the outer gas cushion layer. (**a**) Z velocity at inner surface of the melt; (**b**) Z velocity at outer surface of the melt.

**Figure 6 polymers-14-03559-f006:**
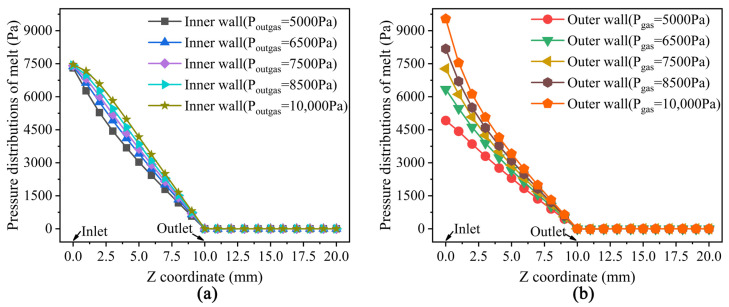
The distribution of melt pressure under different pressure of the outer gas cushion layer. (**a**) Pressure at inner surface of the melt; (**b**) Pressure at outer surface of the melt.

**Figure 7 polymers-14-03559-f007:**
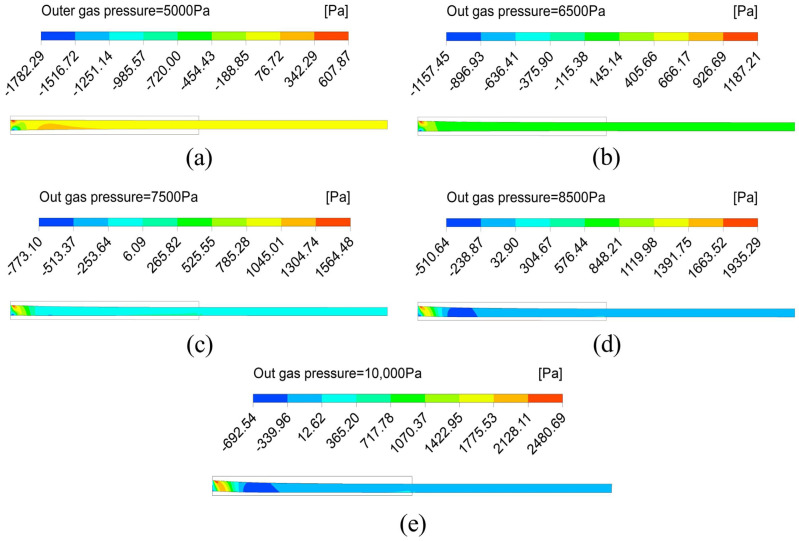
The distribution of melt first normal stress difference under different pressure of the outer gas cushion layer.(**a**) The pressure of the outer gas layer is 5000 Pa; (**b**) The pressure of the outer gas layer is 6500 Pa; (**c**) The pressure of the outer gas layer is 7500 Pa; (**d**) The pressure of the outer gas layer is 8500 Pa; (**e**) The pressure of the outer gas layer is 10,000 Pa.

**Figure 8 polymers-14-03559-f008:**
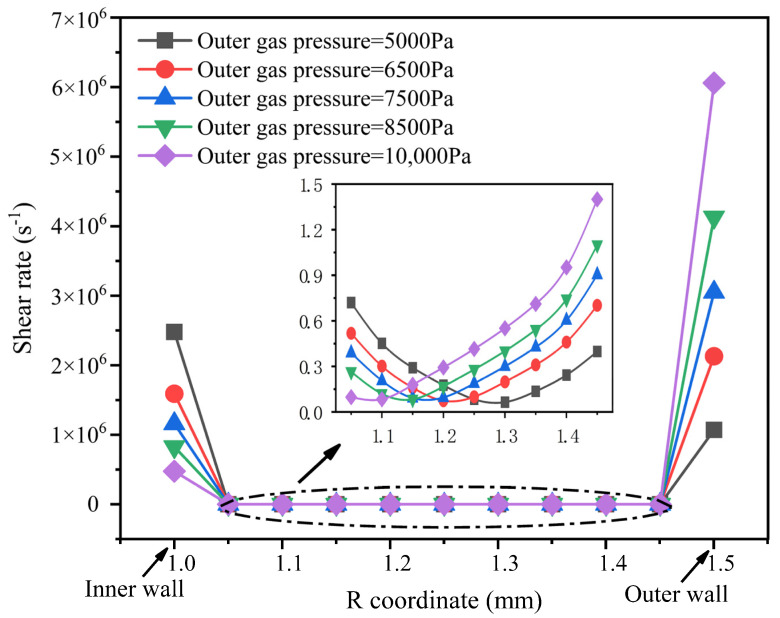
The distribution of melt shear rate field under different pressure of the outer gas cushion layer.

**Figure 9 polymers-14-03559-f009:**
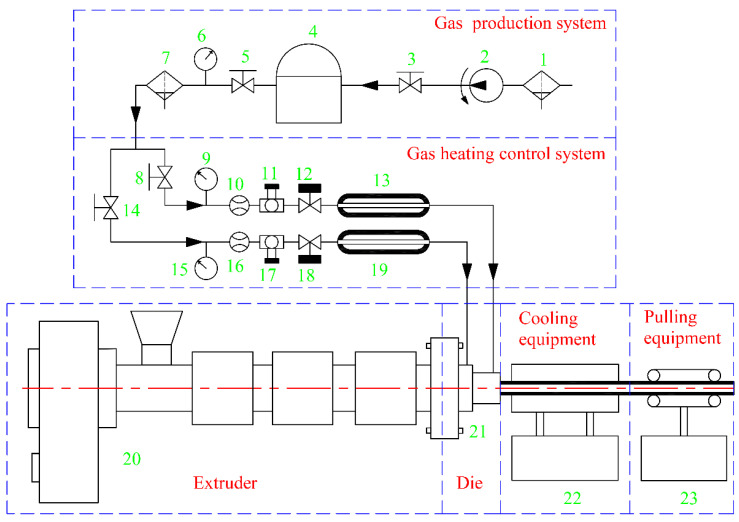
The double-layer gas-assisted extrusion experimental platform. 1,7—Filter; 2—Air compressor; 3,5,8,14—Switch valve; 4—High pressure storage tank; 6,9,15—Pressure gauge; 10,16—Flow meter; 11,17—Pressure regulating valve; 12,18—Flow regulating valve; 13,19—Gas heater; 20—Extruder; 21—Extrusion die; 22—Cooling device; 23—Traction device.

**Figure 10 polymers-14-03559-f010:**
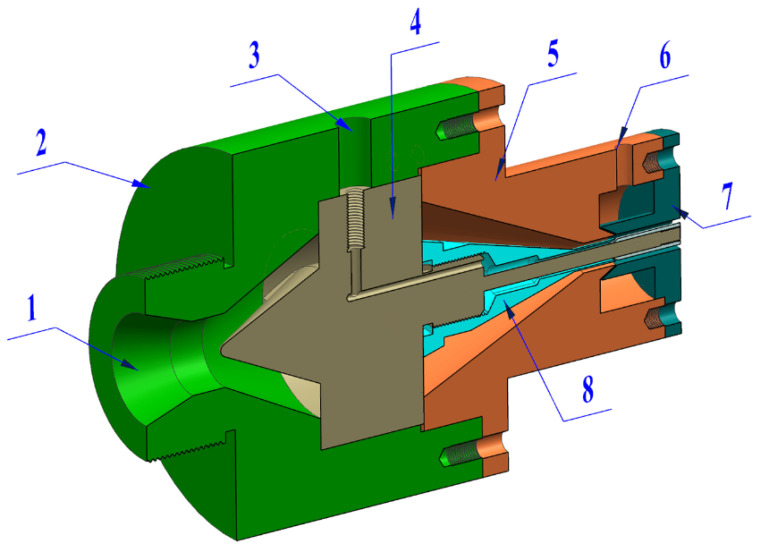
Schematic diagram of the extrusion die. 1—Inlet of melt; 2—Extruder connection module; 3—Inlet of inner gas cushion layer; 4—Front part of the diverter cone; 5—Die connection module; 6—Inlet of outer gas cushion layer; 7—Die; 8—End part of the diverter cone.

**Figure 11 polymers-14-03559-f011:**
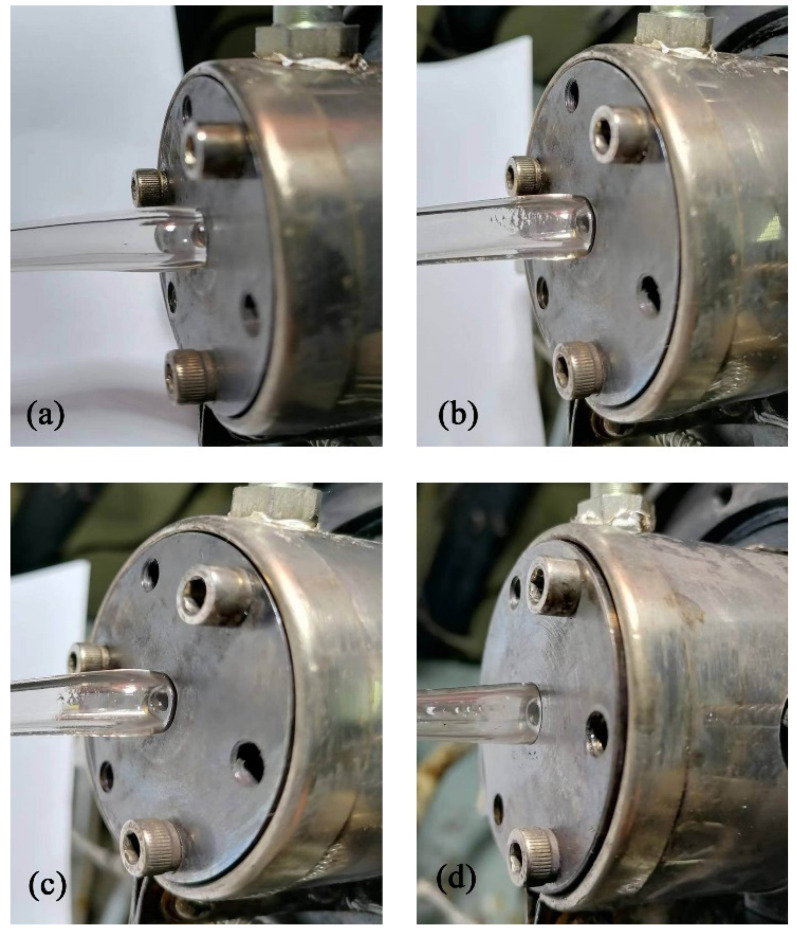
Results of micro-tube extrusion experiment. (**a**) The non-gas-assisted extrusion process; (**b**) Outer gas pressure = 6500 Pa; (**c**) Outer gas pressure = 7500 Pa; (**d**) Outer gas pressure = 10,000 Pa.

**Table 1 polymers-14-03559-t001:** Numerical simulation parameters of the melt and gas.

Equation Parameters	Melt	Gas
$\eta_{1}$/(Pa s)	8823	$2.6\times 10^{-5}$
λ/(s)	0.1	0
ε	0.15	0
ξ	0.44	0
$\eta_{1r}$	0.12	0
$q_{k}/\left(W\cdot m^{-1}\cdot K^{-1}\right)$	0.22	0.037
$C_{V}/\left(J\cdot kg^{-1}\cdot K^{-1}\right)$	1883	0.037

**Table 2 polymers-14-03559-t002:** Experimental conditions.

Experimental Conditions	Gas-Assisted Extrusion	Non-Gas-Assisted Extrusion
Inner gas Pressure/(Pa)	7500	0
Outer gas Pressure/(Pa)	6500\7500\10,000	0
Traction device frequency/(Hz)	4	4
Temperature of the die/(°C)	200	200
Extruder motor frequency/(Hz)	4	4

## Data Availability

Not applicable.

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
