# Peer review of "Effect of Pressure Difference between Inner and Outer Gas Layer on Micro-Tube Deformation during Gas-Assisted Extrusion"

_polymers, 2022, doi:10.3390/polym14173559_

Round 1
Reviewer 1 Report
The manuscript "Effect of pressure difference between inner and outer gas layer on micro-tube deformation during gas-assisted extrusion" is very interesting, new and original. My recommendation is "accept as is".
Author Response
Dear Reviewers:
We appreciate you very much for your positive and constructive comments and suggestions on our manuscript entitled “Effect of pressure difference between inner and outer gas layer on micro-tube deformation during gas-assisted extrusion” (Manuscript ID: polymers-1840108).
Thanks again to the hard work of the editor and reviewer!
Reviewer 2 Report
1. The conclusions should be given with data instead of using general statement only.
2. It is noted that experimental were carried out in parallel to the modelling. Explicit model validation should be carried out to make the models trustworthy.
3. Some grammatical corrections are needed. For example, after each equation "where" instead of "Where" should be used when explaining the variables.
Author Response
Dear Reviewers:
We appreciate you very much for your positive and constructive comments and suggestions .Please see the attachment about the response.
